# Isolated Radial Nerve Palsy in a Newborn Due to a Congenital Myofibroma: A Rare Case of Peripheral Nerve Injury

**DOI:** 10.3390/children11091126

**Published:** 2024-09-15

**Authors:** Serena Chiellino, Viola Fortini, Chiara Castellani, Pierluigi Vasarri

**Affiliations:** 1Paediatric and Neonatologic Unit, Santo Stefano Hospital, 59100 Prato, Italy; pierluigi.vasarri@uslcentro.toscana.it; 2Paediatric Pulmonary Unit, Meyer Children’s Hospital IRCCS, 50139 Florence, Italy; 3Rehabilitation Department, Meyer Children’s Hospital IRCCS, 50139 Florence, Italy; viola.fortini@meyer.it (V.F.); chiara.castellani@meyer.it (C.C.)

**Keywords:** radial nerve palsy, newborn, congenital tumor, myofibroma, tendon transfer

## Abstract

Isolated musculoskeletal infantile myofibroma is a rare tumor of pediatric age. The majority of cases are seen in children under two years old, but it can occur at any age as a painless enlarging mass that involves bone, skin, or soft tissue, typically accompanied by compression symptoms. Perineural involvement is extremely rare in myofibromas. Neurological impairment can occur during infancy but isolated nerve palsy, particularly in peripheral nerves within the upper extremity, is very uncommon. Neonatal radial nerve palsy is a rare entity caused by different conditions. Among these, we mention local tumors affecting peripheral nerves, such as myofibroma. There are few cases described in the literature, which mainly concern adult patients. The authors present a case of congenital isolated radial nerve palsy in a newborn with MF of the right elbow, which resulted in impairment of the wrist and finger extension. Following a six-month monitoring period, the patient underwent surgical treatment to restore function to his right wrist and hand. This involved excising the infiltrated radial nerve segment associated with palliative surgery. Despite the benignity of this lesion, severe nerve damage and perineural involvement may require surgical treatment with nerve resection and reconstruction.

## 1. Introduction

Myofibroma is a rare infiltrating soft-tissue tumor that usually affects infants and young children as either a solitary lesion or in the multicentric form with variable clinical presentations. In the multi-centered form called myofibromatosis, there are multiple benign fibroblastic-myofibroblastic lesions. The most commonly affected body areas are the head, the neck, and the trunk. This form may also present with visceral involvement, which usually has a poor prognosis [1].

Non-traumatic radial nerve palsy (RNP) is a rare disorder resulting from a variety of reasons, but severe RNP is primarily caused by iatrogenic damage during surgical operations and humeral shaft fractures. Local tumors can often compress nerves, mimicking the symptoms of compartment syndrome [2].

The incidence of soft-tissue tumors in newborns is low and is approximately 1/150,000. Infantile myofibromatosis accounts for 35% of the soft-tissue tumors present at birth with an incidence in newborns of 1/400,000 [3].

The etiology of IM is still unknown. Previous studies have supposed that the mutation in the receptor of the platelet-derived growth factor (PDGFRB) causes IM, and NOTCH3 has been identified as an involved gene as well as basic fibroblast growth factor (bFGF) [4].

Local complications are related to the mass effect and compression of the surrounding organs, such as the orbit, the larynx, the nerves, the brachial plexus, and the vertebral canal [1]. Neuropathic symptoms related to compression or infiltration of neurovascular structures are unusual.

Myofibroma often presents with compression symptoms due to its tendency to grow affecting the surrounding organs and tissues and may also affect peripheral nerves. Perineural invasion is very unusual in benign tumors and it is characterized by the invasion of tumoral cells in, around, or through nerve fibers. This mechanism may cause nerve palsy with higher morbidity and worse prognosis. In some cases, the lesion regresses spontaneously in one or two years, with excellent prognosis.

We describe a rare case of a newborn with isolated myofibroma of the right elbow causing perineural invasion of radial nerve. The resulting radial nerve palsy was revealed as irreversible and required removal of the mass and of the damaged nerve tract associated with palliative surgery.

## 2. Case Description

A full-term Italian male infant was born at 40 weeks plus one. His mother, who was a primigest, gave birth vaginally after an eight-hour labor without any complications. Fetal abnormalities were not found during prenatal ultrasound, and prenatal diagnostic tests for genetic disorders and maternal infectious screening were negative. At birth, the newborn weighed 3.770 g, had an Apgar score of 10/10, had a head circumference of 34 cm, and a height of 52 cm. He was exclusively breastfed. The newborn was in good clinical condition but he showed a severe RNP. Right wrist drop was noted immediately after delivery.

The physical examination revealed normal heart sounds and regular breathing movements with normal vital parameters. The abdominal percussion and palpation were normal. A red, violetous mass with erythema was found on the lateral face of the right elbow, and it was related to hypotonia and weakening of the wrist and finger extensors on the upper right limb. Examination of the right upper limb showed that extension of the wrist, thumb, and metacarpophalangeal joints was impossible, and also impairment of prono-supination and complete extension of the ipsilateral elbow was noted; meanwhile, the wrist flexion shoulder movements were preserved. The Moro reflex was asymmetric, while tonic neck reflex was normal. Examination was in keeping with isolated radial nerve palsy. According to grade 0 on the Medical Research Council (MRC) scale (0–5), the strength of extension of the wrist and thumb was completely lost, and grade 2 palsy was indicated by the extension in the metacarpophalangeal (MP) joints II–V. The patient retained full sensory and motor function in the median and ulnar nerve distributions, but lost all sensation in the peripheral radial nerve area.

### 2.1. Diagnostics

Brain ultrasound performed after birth was normal. Blood tests were negative for infectious diseases. Complete blood count, calcium, glycemia, lipid profile, and urine test were normal, and tumor markers were negative. The first diagnostic hypothesis was shaft fracture of the right humerus, but X-ray of the right arm and forearm was negative. Ultrasonography showed a solid, slightly vascularized lesion with a hyperechoic center and a surrounding ring, involving brachial muscle, compatible with intramuscular hematoma, which caused compression of the radial nerve (Figure 1). Studies on nerve conduction revealed that radial motor compound muscle action potential (CMAP) and radial sensory nerve action potential (SNAP) had significantly reduced amplitude distal to the elbow. A watchful, waiting attitude with physiotherapy was performed. The newborn had benefited from passive movement-type physiotherapy sessions and sensorimotor facilitation for upper limb integration. The patient was also fitted with a custom-designed static splint made of thermo-moldable plastic that had perforations, immobilizing only his right wrist and fingers in a neutral position (Figure 2). Thankfully, there were no signs of contractures or stiffness in the wrist or any of the finger joints. At 2 months of life, a second X-ray showed proximal radius deformity, due to a solid mass compression (Figure 3A,B). Magnetic resonance imaging of the right upper limb revealed a solid, hypovascular mass with an area of necrosis infiltrating and involving the radial head, and fatty atrophy in line with proximal extensor muscle denervation alterations. The mass was mildly hypointense on T1-weighted sequences and hyperintense on T2-weighted images, with mostly peripheral post-contrast enhancement (Figure 4A,B). Total body MRI scan made it possible to evaluate the presence of similar lesions in other parts of the body, so to exclude diagnosis of infantile myofibromatosis. CT angiography of the right upper limb was performed to evaluate upper extremity arterial abnormalities and it showed regular patency of arterial vessels, with dislocation of the brachial artery by the mass. The main differential diagnosis included fibrous hamartoma of infancy, juvenile hyaline fibromatosis, infantile fibrosarcoma, or Ewing sarcoma/primitive neuroectodermal tumor. A fine needle biopsy of the lesion confirmed that it was a benign mesenchymal tumor compatible with myofibroma. Genetic testing revealed no familial inheritance. Following physiotherapy, with the same activity improving with taping for the wrist and fingers’ extension, then replaced by a Lycra dynamic orthosis (flexa) (Figure 5) during 3–4 h when awake and stretching exercises, an improvement in the infant’s right arm motility was seen within 4–5 months of birth, but no activation of the wrist and finger extensors was observed. Therefore, it was advised to explore the radial nerve surgically in case of compressive neuropathy.

### 2.2. Operation

Surgical removal of the mass was scheduled at six months of the patient’s life.

The intervention was performed with a volar approach to the radial nerve in the region of the right elbow identifying a circumscribed gray lesion infiltrating the radial nerve and disrupting all his fascicles. The mass had also affected the proximal section of the extensor muscles. It was not possible to perform neurolysis due to internal adhesions of the nerve, so we sectioned the damaged tract but without nerve grafting. All the lesion was removed, but the radial nerve damage was irreparable so we decided to perform a triple tendon transfer technique. Pronator teres (PT) to extensor carpi radialis longus/brevis (ECRL/B), rerouted abductor pollicis longus to palmaris longus (PL), and flexor carpi radial (FCR) to extensor digitorum communis (EDC) and extensor pollicis longus (EPL) were among the transfers performed during the procedure. The Pulvertaft technique (end-to-side: PT to ECRL/B and FCR to EDC and EPL; end-to-end: APL to PL) was followed for all tendon sutures. The wound healed without complications, although the outcome was not very pleasing to the eye.

Following surgery, the right upper extremity was kept immobile for four weeks in an extension position using a plaster splint. In order to preserve his range of motion, the patient was then referred to hand therapy for full-day splinting. The splint could be removed for the physiotherapy treatment with neurosensory activities for active recruitment and sensorimotor facilitation for upper limb integration, mobilization, and stretching and scar care. After 3 months, the static splint, during the awake time, was replaced with a dynamic orthosis in Lycra (flexa) that the patient could use for active play during controlled manipulation. Activities were also carried out to support global psychomotor development. The physiotherapy program was performed with a hands-off daily treatment at home by parents, with excellent level of adherence, and 2/3 times a week hands-on session by pediatric physiotherapy. Histologic examination demonstrated spindle cell proliferation made up of elements organized in fascicles. Tumor cells were spindle-shaped myoid cells, with eosinophilic cytoplasm and tapering nuclei in contact with nerve bundles. The vasculature showed pericyte-like characteristics with thin-walled slit-like vessels. Additionally, there were isolated regions of calcifications and hyalinosis in between the tumor nodules. The lesion cells tested very positive for smooth muscle actin (SMA) by immunohistochemistry, focal positivity was detected for CD99, CD10, calponin, desmin, caldesmon, whereas the S100 protein and CD34 tests were negative. The Ki67 proliferative fraction was about 2%. The diagnosis of MF was confirmed by these histologic findings.

At 6 months postoperatively, the patient acquired optimal recovery and range of motion of the right upper extremity with correct integration of the right upper limb and ability to handle objects differing in size, weight, and shape. Twelve months post-operatively, there was significant improvement of the finger extension and motor power of the wrist. Total body MRI scan demonstrated no tumor recurrence. X-rays of the right arm and forearm showed remodeling of the proximal radius with complete bone healing and no complications involving the growth plate or deformity. The patient presented good recovery of wrist flexion–extension ROM and grasp strength, but he needed to flex his wrist to fully extend his fingers, also a mild lag in thumb metacarpophalangeal extension was observed, which reflects poor recovery of the EPL function.

## 3. Discussion

Our case is an extremely rare case of radial nerve paralysis due to a congenital myofibroma. There are no similar clinical cases previously reported in the literature.

Clinical presentation of our patient was very unusual. Prenatal ultrasound did not allow the identification of the right elbow lesion and this caused a diagnosis delay. After one month, pathology demonstrated an isolated rare soft-tissue tumor, myofibroma. The lesion caused irreversible radial nerve paralysis, due to perineural invasion, with no motor recovery after an observational period of six months. For these reasons, surgical treatment was performed with resection of the mass and of the damaged nerve tract. Associated palliative surgery with tendon transfer procedure and physiotherapy program allowed an early motor recovery of the right wrist and hand function.

Isolated radial nerve palsy is characterized by the loss of extension of the wrist, thumb, and metacarpophalangeal joints, but preservation of the external rotation of the shoulder and elbow movement. Among causes of neonatal radial nerve palsy (NRNP), we mention prolonged intrauterine compression, constriction band, septic arthritis of the shoulder, humerus fractures, infantile cortical hyperostosis (Caffey disease), and prolonged application of arm cuff for blood pressure measurement [5,6,7] (Appendix A). The most frequent mechanism is intrauterine compression, presenting with ecchymosis, erythema, and ulceration on the posterior surface of the arm [6,7].

The main differential diagnosis in the neonatal period is brachial plexus birth palsy which can present with impairment of the elbow flexion, shoulder paralysis, and limitation of the external rotation of the upper limb [8]. Brachial plexus birth palsy can be distinguished from congenital radial nerve palsies with evaluation of the Moro which is normal in congenital radial nerve palsies, and asymmetric tonic neck reflexes, normal in an isolated peripheral radial nerve injury as well as symptoms of Horner’s syndrome.

The true incidence of this condition remains unknown, mainly due to its spontaneous recovery [9].

Determining if a specific anatomic etiology, decreased fetal activity for oligohydramnios, or another source of external compression, also during birth, exists in a newborn with isolated radial nerve palsy is crucial [10,11].

It is still debatable how to clinically treat nerve palsies with unclear causes [12,13,14,15]. According to earlier research [9], the observational time before surgical intervention for these lesions might vary from one month from the beginning of symptoms to seven months in the absence of any signs of functional improvement. Since 66% of patients will exhibit motor recovery within a month of clinical start, an observational period is advised in cases where spontaneous remission may occur [16].

Diagnostic imaging tests are essential for evaluating the extent of the disease, characterizing the lesions, assessing visceral involvement, and following up the progression of the lesions. Ultrasound, MRI, and eventually high-quality CTA of the upper extremities are recommended to guide the diagnosis and surgical planning. Sometimes, a low radiation dose CTA allows display of both vascular and musculoskeletal structures, in order to evaluate traumatic etiology [17].

Long periods of observation—typically longer than seven months—without signs of nerve healing are linked to worse outcomes [13,14,18].

When there is no sign of improvement after seven months, prior research has shown that surgical intervention, which should involve both internal neurolysis and external decompression, improves motor recovery [13,14,18,19]. When internal adhesions prevent neurolysis or when scarring covers more than 75% of the nerve, as it did in our instance, nerve grafting and removal of the scarred segment have demonstrated good functional recovery. Myofibromas most frequently affect pediatric patients and it is often multicentric in the pediatric population [20,21,22].

In order to avoid a misdiagnosis, clinical history and diagnostic imaging must be carefully studied. Clinical presentations can guide physicians in performing adequate management. Congenital radial nerve palsy, without anatomic abnormalities, is usually managed with observation and physical therapy because most cases spontaneously resolve with complete recovery and no sequelae without intervention.

In the case of extensive damaged nerve segments, surgical treatment is needed. After nerve resection is advisable to proceed with nerve reconstruction. The nerve gap may be reconstructed using autologous cabled grafts. If nerve grafting is not feasible, palliative surgery such as tendon transfer techniques can be utilized.

Our work is focused on the rarity of the disease and the efficacy of early surgical treatment. In our case, despite the benignity of the lesion, considering the lack of clinical remission and the structural damage of both nerve and radius, we decided to proceed with the removal of the mass and resection of the damaged radial nerve tract. Another important consideration is that during infancy, normal hand function is essential to acquire a correct psychomotor development, so we preferred early nerve damage repair with tendon transfers to allow the young patient an early use of the right hand and therefore promote a correct integration of the injured limb. In order to achieve these objectives, it is crucial to receive timely physiotherapy treatment following surgery.

## 4. Conclusions

The main purpose of our work is to describe the rarity of congenital radial nerve palsy due to a tumor compression and provide an effective clinical strategy for early diagnosis and consequently good response to medical and surgical treatment.

According to our experience, early surgical intervention can improve recovery in cases of congenital solitary nerve palsy with anatomic abnormalities. Given the rarity of this condition, further studies are required to better evaluate the risk/benefit ratio and outcomes of surgical intervention and application of tendon transfer procedure in this pediatric population.

## Figures and Tables

**Figure 1 children-11-01126-f001:**
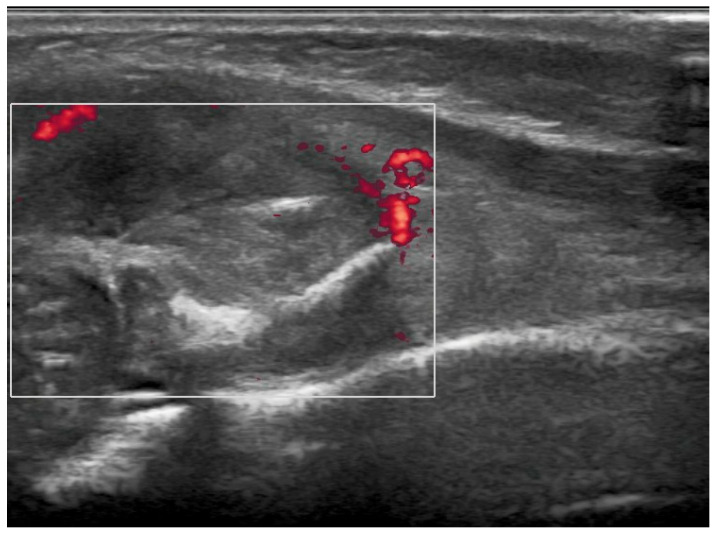
Preoperative ultrasound imaging of right elbow: soft tissue lesion on the anterolateral surface of the right forearm characterized by a nonhomogeneous echotexture and hyperechogenic spots and thin hyperechogenic lamella in its context. Intralesional vascularization is minimal. The surface of the cortex of the proximal radial metaphysis facing the mass is irregular. There is no evidence of chondro-epiphyseal detachment.

**Figure 2 children-11-01126-f002:**
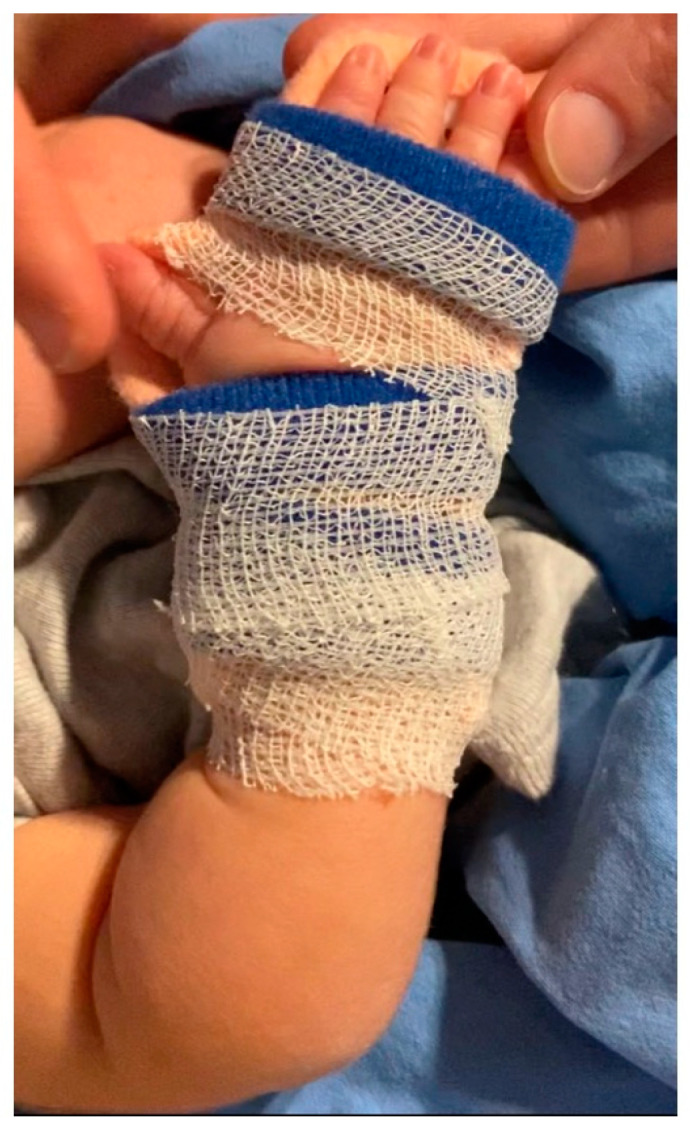
Static splint immobilizing right wrist and fingers in neutral position.

**Figure 3 children-11-01126-f003:**
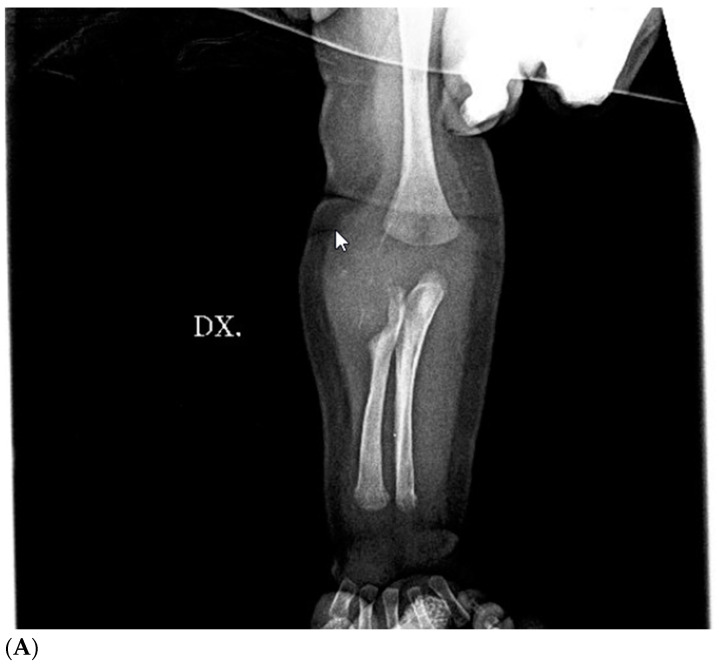
Preoperative X-ray of right arm and forearm: large mass on the anterolateral surface of the right forearm near the elbow crease compressing (**A**) and eroding (**B**) the head of the right radio.

**Figure 4 children-11-01126-f004:**
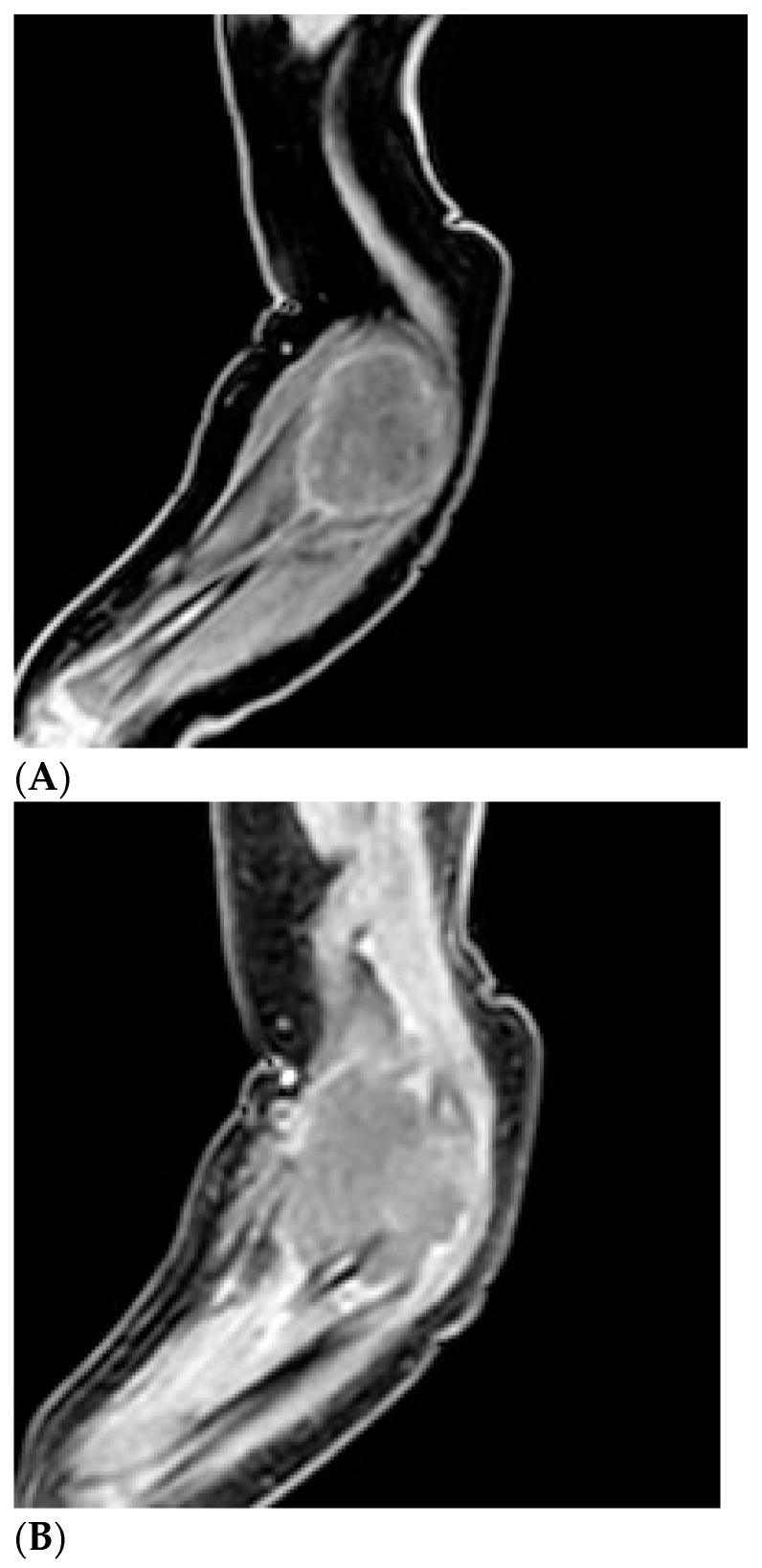
Preoperative magnetic resonance imaging: presence of an expansive lesion of the anterolateral compartment of the right elbow with the myotendinous junction of the brachialis muscle as its epicenter, until it incorporates the proximal radius, partially eroding it. This lesion has welldefined margins with alteration of the signal of the surrounding soft tissues: hypointense on T1 (**A**) and hyperintense on T2 (**B**) and predominantly peripheral lesion impregnation.

**Figure 5 children-11-01126-f005:**
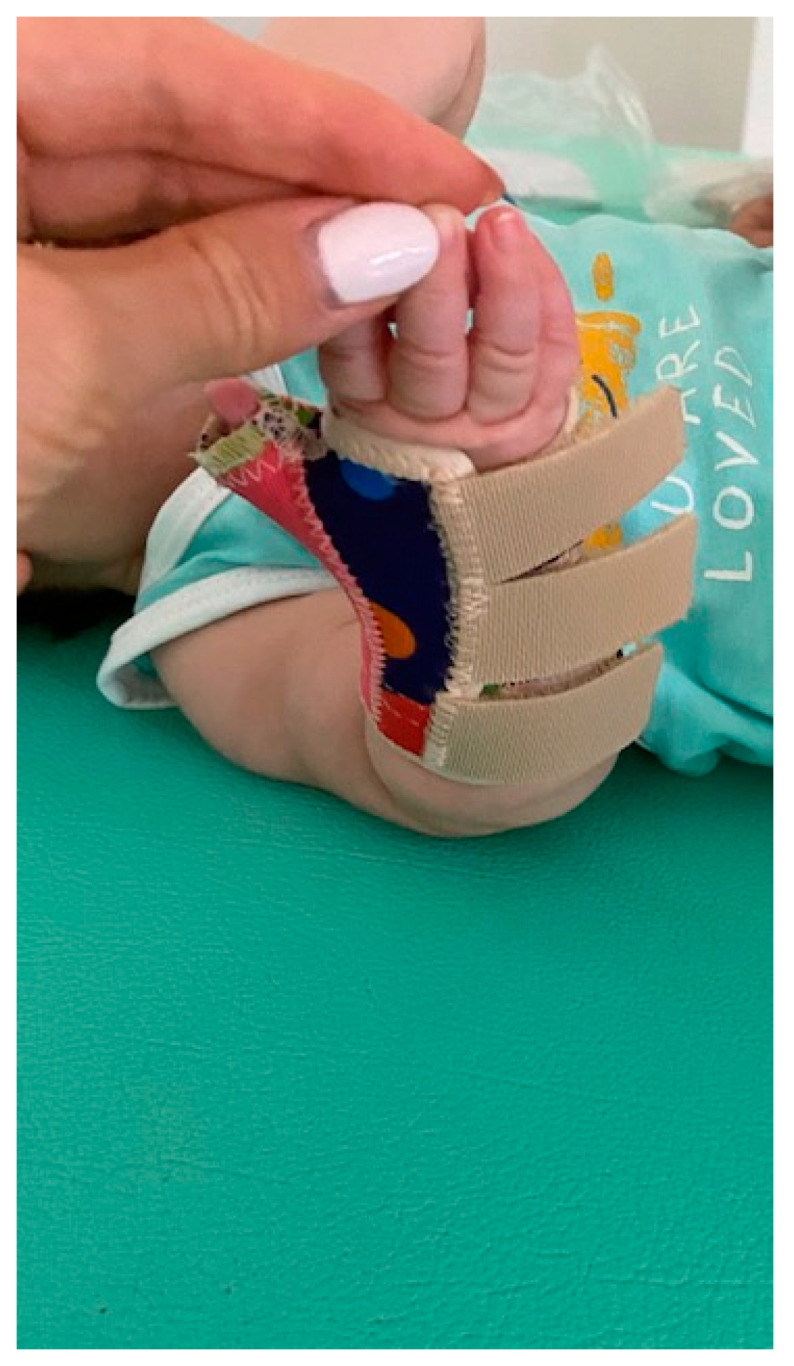
Lycra dynamic orthosis flexa applied on the patient’s right wrist, thumb, and hand excluding the other fingers.

## Data Availability

The original contributions presented in the study are included in the article/Appendix A, further inquiries can be directed to the corresponding author/s.

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
