# Peer review of "Isolated Radial Nerve Palsy in a Newborn Due to a Congenital Myofibroma: A Rare Case of Peripheral Nerve Injury"

_children, 2024, doi:10.3390/children11091126_

Round 1
Reviewer 1 Report
Comments and Suggestions for Authors
Dear authors,
It is an honour for me to read and comment on your article. This is a very interesting case report regarding an indeed rare pathology.
Your article may be and deserves to be improved and published.
My suggestions are as following:
-the abstract is focus on the myofibroma, without mentioning the radial nerve plasy; and you should do that, as the main idea of the article is the association between the two;
-after that, you focused the report on the radial nerve palsy, which is correct since this is part of the problem, but you must discuss this issue in the context of the cause. In my opinion, you should also focus on mass effect causing nerve palsy, since this was here the cause. And a little more on myofibroma.
I sugggest the following:
Popa Ș, Apostol D, Bîcă O, Benchia D, Sârbu I, Ciongradi CI. Prenatally Diagnosed Infantile Myofibroma of Sartorius Muscle—A Differential for Soft Tissue Masses in Early Infancy. Diagnostics. 2021; 11(12):2389. https://doi.org/10.3390/diagnostics11122389
Schmidt, D. Fibrous tumors and tumor-like lesions of childhood: Diagnosis, differential diagnosis, and prognosis. Curr. Top. Pathol. 1995, 89, 175–191
-it is useful to add some pathology pictures if you have.
-discussion section must be revised since there are large paragraphs mixing the ideas (eg: first one is talking about infantile myofibroma and also about the differentials in RNP)
-please check again the text for the discussions regarding the adult pathology; in my opinion, they are to extensive since the case was about an infant (eg: lines 185-195 - trauma: open reduction and internal plate fixation, intramedullary nailing, elbow arthroplasties - this is not the case in newborn and infant pathology; the same, "Saturday night palsy")
-it will be useful to add a table with the articles reporting RNP in newborn and infant, since this is indeed a rare pathology
-be careful at the CT indication, since we are talking about infants
-please discuss the surgical option in the discussions section and not in the conclusions
Comments on the Quality of English Language
-English should be revised
Author Response
- I agree with these comments so I've modified abstract according your suggestions
- I've revised introduction by elaborating and extending general information
-I've revised entire discussion. The paragraphs concerning adult RNP and iatrogenic RNP have been deleted (es lines 185-195). In the first part of discussion I've explained the importance of our case report. Also the paragraph concerning clinical and surgical management of our patients has been added to discussion and deleted from conclusions.
-Grammatical rules have been revised and also division into paragraphs has been applied
- I've added a table concerning neonatal upper limb palsy ( causes, risk factors and differential diagnosis)
- unfortunately we have no pictures of pathology or surgical intervention
Reviewer 2 Report
Comments and Suggestions for Authors
The abstract contains too much general information. You should move the general information to the Introduction, it may be useful to mention more facts and their importance in the abstract.
In the introduction, there is very little general information. Extend the introduction by elaborating and extending the general information.
The plagiarism rate is too high. Please make the necessary adjustments. Originalise your article
Your case narrative is detailed and quite comprehensive. All important details are mentioned. However, the grammatical rules were not obeyed. Please divide the article appropriately into paragraphs.
In the first part of the discussion, you should mention the importance of your case and your important contribution to the literature. It is better to mention and give general information in the Introduction section.

Comments on the Quality of English LanguageThe grammatical rules were not obeyed. Please divide the article appropriately into paragraphs.
Author Response
- I agree with these comments so I've modified abstract according your suggestions
- I've revised introduction by elaborating and extending general information
-I've revised entire discussion. The paragraphs concerning adult RNP and iatrogenic RNP have been deleted (es lines 185-195). In the first part of discussion I've explained the importance of our case report. Also the paragraph concerning clinical and surgical management of our patients has been added to discussion and deleted from conclusions.
-Grammatical rules and plagiarism have been revised and also division into paragraphs has been applied